# An Interactive Self-Learning Game and Evolutionary Approach Based on Non-Cooperative Equilibrium

**Yan Li, Mengyu Zhao, Huazhi Zhang, Fuling Yang and Suyu Wang \***

School of Mechanical Electronic & Information Engineering, China University of Mining and Technology-Beijing, Beijing 100083, China; 201572@cumtb.edu.cn (Y.L.); zmy@student.cumtb.edu.cn (M.Z.); zhz@student.cumtb.edu.cn (H.Z.); flyang@cumtb.edu.cn (F.Y.)

**\*** Correspondence: wsy@cumtb.edu.cn; Tel.: +86-010-62331083

**Abstract:** Most current studies on multi-agent evolution based on deep learning take a cooperative equilibrium strategy, while interactive self-learning is not always considered. An interactive self-learning game and evolution method based on non-cooperative equilibrium (ISGE-NCE) is proposed to take the benefits of both game theory and interactive learning for multi-agent confrontation evolution. A generative adversarial network (GAN) is designed combining with multi-agent interactive self-learning, and the non-cooperative equilibrium strategy is well adopted within the framework of interactive self-learning, aiming for high evolution efficiency and interest. For assessment, three typical multi-agent confrontation experiments are designed and conducted. The results show that, first, in terms of training speed, the ISGE-NCE produces a training convergence rate of at least 46.3% higher than that of the method without considering interactive self-learning. Second, the evolution rate of the interference and detection agents reaches 60% and 80%, respectively, after training by using our method. In the three different experiment scenarios, compared with the DDPG, our ISGE-NCE method improves the multi-agent evolution effectiveness by 43.4%, 50%, and 20%, respectively, with low training costs. The performances demonstrate the significant superiority of our ISGE-NCE method in swarm intelligence.

**Keywords:** non-cooperative equilibrium; interactive self-learning; generative adversarial; game evolution; multi-agent confrontation

## 1. Introduction

Deep learning has received high attention from research institutions and industries since its emergence in 2006. In recent years, deep reinforcement learning (DRL) has gradually become a research focus and the development trend in the field of artificial intelligence and was selected by MIT as one of the 10 breakthrough technologies in 2017 [1]. Various deep learning methods such as Dagger [2], Deep Q-learning [3], DRL [4] and DNQ [5] are publicized. Among them, the Deep Deterministic Policy Gradient (DDPG) is widely used because of its excellent ability to observe and execute actions instantly in terms of individual intelligence [6], such as for the robotic arms to achieve high precise actions [7], for the Autonomous Underwater Vehicles (AUVs) to patrol intelligently [8]. However, DDPG has problems such as behavioral convergence failure and low training efficiency when dealing with multi-agent environment behavior problems [9,10].

Accordingly, multi-agent intelligent learning has been intensively studied and conceived as deep learning develops. For instance, Malysheva A. et al. [11] proposed a new MAGNet method for multi-agent reinforcement learning. Based on a weight agnostic neural networks (WANNs) methodology, an automated searching neural net architecture strategy was proposed that can perform various tasks such as identifying zero-day attacks [12]. Sheikh, HU et al. [13] proposed the DE-MADDPG method, which can achieve better multi-agent learning by coordinating local and global rewards. Li, SH. et al. [14] proposed

the M3DDPG method, which can be better adapted to the multi-agent environment. All of the above multi-agent intelligent learning methods have the function of interactive learning, which proves that intelligent learning in multi-agents can achieve self-learning and evolution for the individuals; effectively, it is advanced in learning ability compared with the methods such as DDPG, which concerns only self-learning as an agent [15]. However, these methods lack the support of game theory.

It has to be noticed that multi-agent interactive learning is a well-known characteristic of game evolution. In recent years, many scholars have introduced the idea of "game theory" into deep learning. For example, in the battlefield of cyberspace, an offense–defense evolutionary game model [16] as well as an exploration–detection game model [17], was established to constantly improve cognition and decision optimization by interactive learning, such as to enhance cyber defense capability. In terms of the communication battlefield, the establishment of a dynamic game model [18], which allows intelligent jammers and intelligent anti-interference to be trained in interactive game confrontation, can effectively improve the intelligence of the multi-agent. In summary, it is obvious that combining game evolutionary with a multi-agent interactive learning framework can effectively improve the training effect. However, these studies did not introduce game strategies.

There are some references that apply game theory strategies to deep learning appropriately. In [19], an iterative algorithm was proposed in order to provide existence and convergence conditions under which the buses reach a suitable equilibrium. However, this study did not use an interactive learning framework. Shi, D. et al. [20] proposed a mean-field game (MFG)-guided deep reinforcement learning (DRL) approach for task placement in collaborative MEC, which helps servers to make instant task placement decisions and significantly reduces the average service latency. Zhang, HD et al. [21] proposed a learning method that can measure the collaborative information between multiple agents, using a cooperative approach to solve the problem of individual robot capability limitation. Liu, M et al. [22] proposed a heuristic imprecise probabilistic based interaction decision algorithm, HIDS, i.e., it utilizes the multidimensional semantic relevance among observation, tasks and agents, such that agents can continuously improve cognition and optimize decision-making by learning interactive records. However, the mentioned game strategies above are mainly based on cooperative parallelism while not considering the non-cooperative equilibrium between the two parties of the game.

Non-cooperative game equilibrium is a critical phase in game theory. Shi, D. et al. [23] presented a concept that both sides of the game, regardless of the game process, would eventually reach non-cooperative equilibrium. In [24], the conditions to achieve a locally asymptotically stable Nash equilibrium (NE) and the necessary conditions to achieve an evolutionary stable strategy (ESS) were derived. Yan, PY et. al. [25] developed an iterative algorithm that solves the production decision and scheduling problem based on a non-cooperative equilibrium strategy with cyclic bi-value graphs. Zhu et. al. [26] combined, with the Nash equilibrium strategy, an online minimaxQ network learning algorithm proposed for network training with observed values. Nevertheless, none of the non-cooperative game equilibrium-based deep learning methods collected above had introduced interactive self-learning simultaneously. This paper solved the problems of poor effect and slow learning speed of DDPG on multi-agent learning. Furthermore, the concept of "game evolution" is successfully introduced into the interactive self-learning framework, and the multi-agent interactive self-learning game and evolution method was designed. Finally, the result of agent training achieved the effect of non-cooperative equilibrium

In this paper, an interactive self-learning game and evolution method (ISGE-NCE) based on non-cooperative game equilibrium is designed to realize multi-agent intelligent gaming and evolution. The rest of this paper is arranged as follows: Section 2 introduces the environment modeling of game evolution and the fundamentals of deep learning by taking the DDPG individual intelligent learning method for instance. Section 3 presents

the detailed design of the proposed ISGE-NCE method. The generative adversarial network is first formed to achieve fast individual classification in the environment. The multi-agent interactive learning method is then designed based on distributed non-cooperative equilibrium strategies to realize interactive self-learning games and evolution. Section 4 illustrates the performance of the proposed ISGE-NCE for multi-agent evolution in three different experiment scenarios for effective evaluations. Finally, the work of this article is summarized and discussed in Section 5.

## 2. Related Work

This section introduces the game evolutionary environment modeling and the DDPG learning methods.

### 2.1. Environmental Modeling of Game Evolution

In agent game confrontation, it is difficult to pre-design behaviors for an agent when the environment is complex and changes with time. Traditional behavior planning cannot meet the requirements of game evolution in confrontation, and the strategy of an agent should react differently and adaptively with the change of environment. Deep learning allows agents to learn new strategies online, which helps to gradually improve the performance of the agent and realize the evolution of strategies through the interaction between the agent and the environment. The process can be described as: in a certain state space and action space, training each agent is trained by maximizing the discount reward of the agent. This process is a Markov decision process.

In the Markov decision process, the state at the next moment is determined only by the current decision, independent of all previous states. In addition, the state of each moment corresponds to an action, and the state of the next moment is determined by the action of the current moment. The pros and cons of that state are represented by a determined value which determines the expectation of future returns. Therefore, the concept of value function is introduced in the process of deep learning to represent the value that the state has at the current moment, and the definition equation is as follows:

$$V^\pi(s) = E[R_{t+1} + \lambda V^\pi(S_{t+1})|S_t = s] \tag{1}$$

where $\pi$ is the strategy selection, $V^\pi(s)$ denotes the state value function, $E$ is the expectation, $\lambda$ is the discount factor, $R_{t+1}$ is the reward at moment $t+1$, and $S_{t+1}$ is the state at moment $t+1$. From Equation (1), it can be seen that the current state value is determined by the current reward and the next moment value, which is the basic form of Bellman's equation. The Bellman equation is solved by iteration.

Action value is more reasonable than state value, and the optimal decision action can be selected based on action value. The action value function $Q^\pi(s,a)$ in game evolution is expressed as the cumulative reward obtained after performing action $a$ in state $s$ at the current moment. Its Bellman equation is as follows.

$$Q^\pi(s,a) = E[R_{t+1} + \lambda Q^\pi(S_{t+1}, A_{t+1})|S_t = s, A_t = a] \tag{2}$$

Game evolution is the process of finding the optimal solution to the Bellman equation. From Equation (2), it can be seen that the optimal strategy is to solve the optimal action value function, and the optimal action value function is the maximum value of the action value function among all strategies. The process can be expressed as iterating over $Q^\pi(s,a)$ until convergence, at which point the action corresponding to the optimal action value is chosen as the optimal policy.

### 2.2. DDPG Learning Method

DDPG is a deep learning model with AC (Actor-Critic) architecture, which for any agent, there is a corresponding policy network together with a critic network. At a certain moment $t$, the policy network generates successive actions $A_t$ based on the state $S_t$ in

the environment; after that, $A_t$ interacts with the environment to generate the state $S_{t+1}$ at the next moment. In addition, the critic network will obtain the evaluation value $Q$ of the decision process of the policy network at moment $t$ based on $S_t$ and $A_t$. The strategies and evaluations of different agents in DDPG do not affect each other, that is, each agent's strategies and evaluations are determined only according to its own actions and states.

The policy network is iteratively updated by the parameter $\theta^u$ while $A_t$ is selected based on $S_t$ to produce the current reward $R_t$ and the next state $S_{t+1}$ via environmental interaction. The objective policy network selects the optimal next action $A_{t+1}$ by $S_{t+1}$ sampled in the empirical replay pool. Its network parameters $\theta^{u\prime}$ are periodically replicated from $\theta^u$. The critic network is updated by iterations of the value network parameters $\theta^Q$ and the current $Q$value, i.e., $Q(S_t, A_t, \theta^Q)$, is calculated. The target $Q$value is calculated by $y_i = R_t + \gamma Q'(S_{t+1}, A_{t+1}, \theta^{Q\prime})$ where the computation of $Q'(S_{t+1}, A_{t+1}, \theta^{Q\prime})$ is taken responsible by the target critic network. The network parameters $\theta^{Q\prime}$ are periodically replicated from $\theta^Q$. The loss function of the evaluation network is

$$L(\theta^Q) = \frac{1}{K} \sum_{t=1}^{K} (y_t - Q(s_t, a_t, \theta^Q))^2 \tag{3}$$

The goal of the critic network is to minimize the loss $L(\theta^Q)$, since a smaller $L(\theta^Q)$ represents a more accurate value given for states and actions. Therefore, the parameter $\theta^Q$ of the evaluation network is updated by the gradient descent method, and the equation is shown as (4).

$$\theta^Q = \theta^Q - \eta \frac{\partial}{\partial \theta^Q} L(\theta^Q) \tag{4}$$

where $\eta$ denotes the learning rate.

For policy networks, the larger the feedback $Q$ value obtained, the smaller the loss of Actor would be. The objective function equation is as follows.

$$J(\theta^u) = \frac{1}{K} \sum_{t=1}^{K} Q(s_t, a_t, \theta^u) \tag{5}$$

where $a_t = \pi_{\theta^u}(s_t)$

Since the strategy is deterministic, the gradient of Equation (5) can be expressed as

$$\nabla_{\theta^u} J(\theta^u) = \frac{1}{K} \sum_{t=1}^{K} \nabla_{\theta^u} \pi(s_t, \theta^u) \nabla Q(s_t, a_t, \theta^Q) \tag{6}$$

The parameters $\theta^u$ of the policy network are updated using the gradient ascent method, as shown by Equation (7).

$$\theta^u = \theta^u + \eta \frac{\partial}{\partial \theta^u} J(\theta^u) \tag{7}$$

Since the network update equation of the DDPG learning method does not consider the states and actions of other agents, it cannot achieve interactive self-learning for game evolution. For complex multi-agent environments, the training results of the DDPG learning method are difficult to converge; it is also difficult to achieve a non-cooperative equilibrium. In order to solve this problem, this paper proposes an ISGE-NCE method. Generative adversarial networks and group interactive learning framework are designed to overcome the shortcomings of DDPG method.

## 3. System Design

This section begins with the composition of the evolution system, which is followed by the specifications of the two main components, that is, the framework of the generative adversarial network, and the framework of multi-agent interaction learning, separately.

### 3.1. System Composition

As shown in Figure 1, the interactive self-learning game and evolution system based on non-cooperative equilibrium mainly consists of two parts: generative adversarial network and multi-agent interaction learning. ISGE-NCE system overcomes the incompetence of DDPG 's long training time and poor effect on multi-agent learning. The generative adversarial network part realizes the rapid identification of monomer categories to improves the training speed. In addition, the interactive learning framework part enhances training capacity.

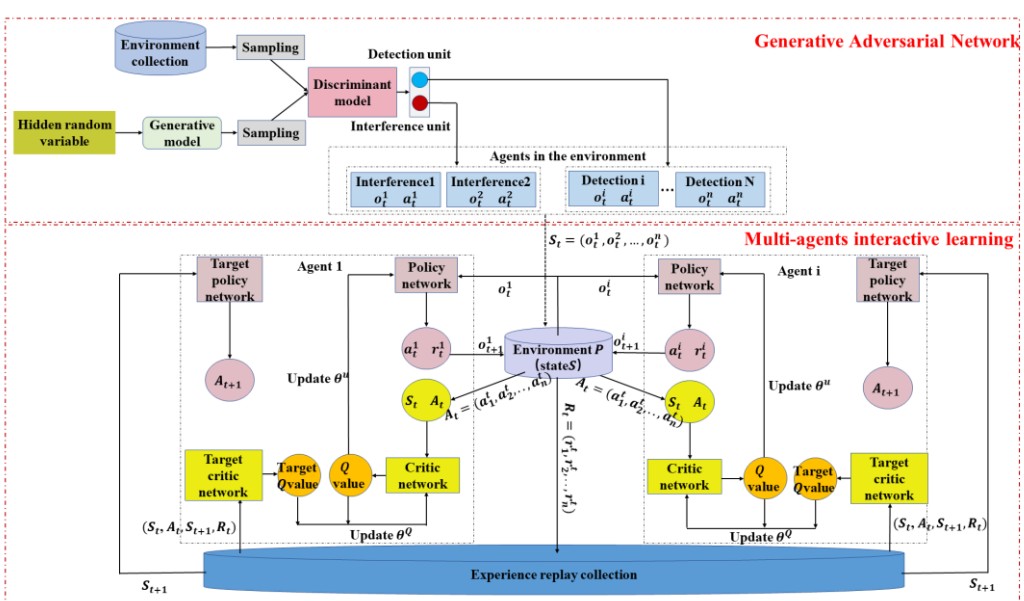

**Figure 1.** Interactive self-learning game and evolutionary system based on non-cooperative equilibrium.

Generative adversarial networks can quickly classify the individual under certain evolution environment. There are agents different in category in the environment set, and the generative model continuously generates new agents by introducing hidden random variables. The discriminant model takes the form of a centralized sampling to distinguish between concrete classes of both sides. Both the generative and discriminative models are updated using back propagation to achieve a stable equilibrium. The generative adversarial network lays the foundation for multi-agent interactive learning and helps to improve the interactive recognition capability and classification speed of the system.

Based on the common rules, the multi-agent interactive learning has an additional parameter sharing mechanism compared with DDPG learning. In addition, the critic network would consider the state and action parameters of other agents, which enhances the interactive learning ability of the agents and speed up the training process as well. Besides, the game evolution of the multi-agent would promote the learning efforts and improve the learning efficiency. Training results reflect non-cooperative equilibrium.

### 3.2. Generative Adversarial Network

The Generative Adversarial Network (GAN) obeys the two-person zero-sum game theory [27,28]. The sum of the interests of the two game participants is a constant, and one of the two players is the generative model (G) and the other is the discriminative model D), and the two parties have different functions.

The framework of the generative adversarial network is shown in Figure 2. The generation model is a sample generator which produces a realistic sample by passing random noise through a multilayer perceptron. The discriminant model is a binary classifier to identify whether the input sample is true or false. The generative model together with the

discriminative model form a generative adversarial network, which improves its own discriminative ability to achieve accurate agent/individual classification by simulating the multi-agent confrontation game and evolution regularly.

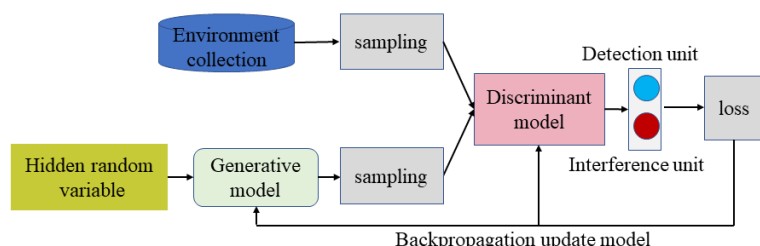

**Figure 2.** Framework of the generative adversarial network.

Optimized discriminant model D.

$$max_D V(D,G) = E_{x \sim pdata(x)}[log(D(x))] + E_{z \sim P_z(z)}[log(1 - D(G(z)))] \tag{8}$$

where $max_D V(D,G)$ denotes maximizing the discrimination of $D$, $E$ is the mathematical expectation, $D(x)$ is the discriminant model function, $D(G(z))$ is the generative model function, and $pdata(x)$ is the environmental sample.

Optimized generative model G

$$min_G V(D,G) = E_{z \sim P_z(z)}[log(1 - D(G(z)))] \tag{9}$$

where $min_G V(D,G)$ denotes minimizing the discrimination of $G$ and $P_z(z)$ is the noise sample.

The generative adversarial network can be subdivided into two steps during the same round of gradient inversion: the discriminative model training first, followed by the generative model training. The training flow chart is shown in Figure 3. During the training of the discriminative model, the surrounding agents are classified to be 0 and 1, while the interference sources are corresponding to 0 and detection sources are corresponding to 1.

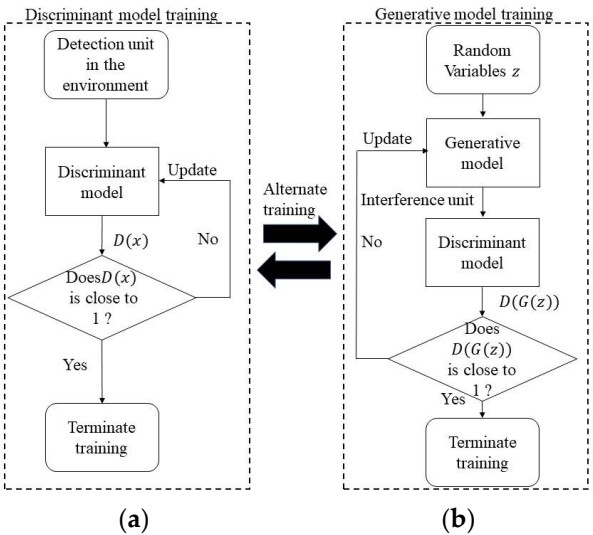

(a)           (b)

**Figure 3.** Generative adversarial network training flow chart: (**a**) discriminant model training; (**b**) generate model training.

The discriminant model samples the set of environments to generate a score $D(x)$, which is back-propagated and updated according to a loss function composed of the score $D(x)$. When training the generative model, the generative model and the discriminative model are considered as a whole, and the overall output score $D(G(z))$ close to 1 is the

goal of optimization. Where the generative model randomly generates different classes of agents through random variables. The generative model is updated in the same way as the discriminant model, with backpropagation updates based on a loss function consisting of score $D(G(z))$. The pseudo-code for generating adversarial networks is shown in Algorithm 1.

---

**Algorithm 1. Generating adversarial networks pseudocode**.

Minibatch stochastic gradient descent training of generative adversarial nets. The number of steps to apply to the discriminator, $k$, is a hyperparameter. $k = 1$ was used, the least expensive option in our experiments.

1    **for** number of training iterations do
2    **for** k steps do
3    Sample minibatch of m noise samples $\{z^{(1)}, \dots, z^{(m)}\}$ from noise prior $p_g(z)$.
4    Sample minibatch of m samples $\{x^{(1)}, \dots, x^{(m)}\}$ from data generating distribution $p_{data}(x)$.
5    Update the discriminator by ascending its stochastic gradient:
6    **end for**
7    Sample minibatch of m noise samples $\{z^{(1)}, \dots, z^{(m)}\}$ from noise prior $p_g(z)$
8    Update the generator by descending its stochastic gradient:
$$min_G V(D, G) = E_{z \sim P_z(z)}[log(1 - D(G(z)))]$$
9    **end for**

The gradient-based updates can use any standard gradient-based learning rule. Momentum was used in our experiments.

---

The training results of the generative adversarial network are shown in Figure 4. As can be seen in Figure 4a, there is no obvious confrontation process between the generative model and the discriminant model at the beginning of training when the number of training is 1300, and their loss values do not change significantly. As shown in Figure 4b, the loss values of the generative and discriminant models tend to stabilize as the number of training times increases. This indicates that the discriminant model can discriminate detection from interference in a simulation game adversarial environment. Generative adversarial networks can effectively improve the adversarial ability of an agent, with fast training, simple model, and high discriminant accuracy after training.

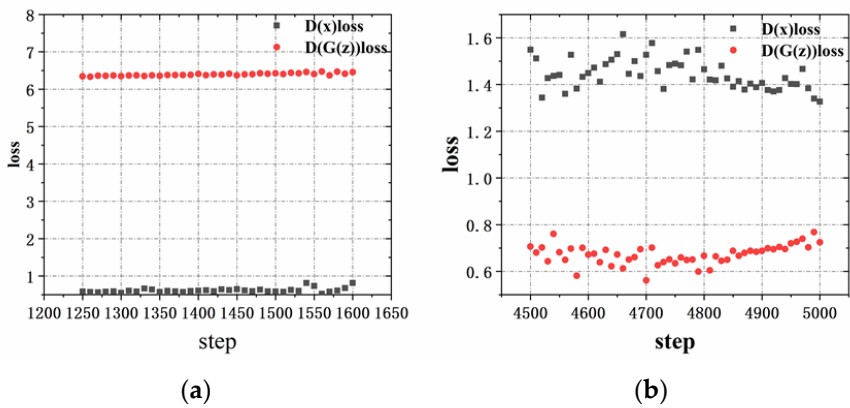

**(a)**        **(b)**

**Figure 4.** Generative adversarial networks training data: (**a**) 1300 training sessions; (**b**) 4500 training sessions.

### *3.3. Multi-Agent Interactive Learning*

3.3.1. Multi-Agent Interactive Network

The individual deep learning algorithm cannot learn interactively because it does not effectively use the global information about the actions and states of other agents. For

game adversarial training of multi-agent, the game adversarial interactive training problem is difficult to solve. Therefore, for the intelligent learning problem of multi-agent, a network framework based on DDPG can form a multi-agent interactive learning algorithm by assigning a set of DDPG networks to each agent. The structure of the multi-agent interactive learning network is shown in Figure 5, which uses a centralized critic network decentralized policy network. The critic network knows not only its own state and action, but also the state and action of other agents when performing evaluation to calculate the value $Q$, and then realize the information interaction of the multi-agent. The multi-agent interactive learning utilizes distributed execution strategy, which allows the agents to set different reward functions according to the task, thus effectively solving the multi-agent non-cooperative strategy execution problem.

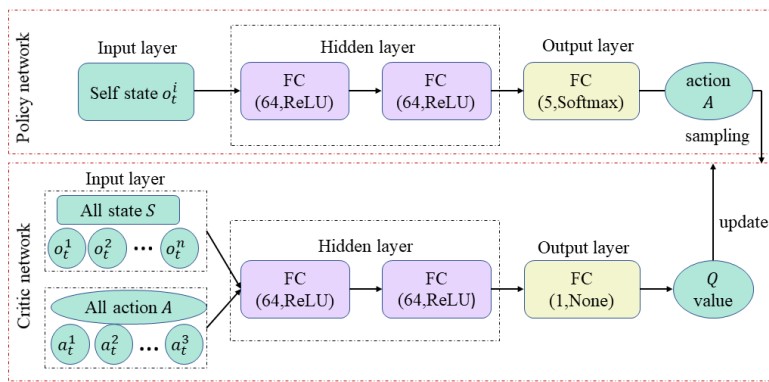

**Figure 5.** Multi-agent interactive learning network map.

In Figure 5, the policy network consists of an input layer, two hidden layers and an output layer. The hidden layer is represented by FC (Fully connection), which is defined by Equation (10).

$$y = \sigma(Wx + b) \tag{10}$$

In Equation (10), x is the input of the network, y represents the output of the network, W denotes the parameter matrix corresponding to the learning of the connection layer $W = (w_1, w_2, \ldots, w_n)$, $\sigma$ is the activation function, and b is the bias vector to be learned by the network.

Each hidden layer has 64 neurons, and ReLU is used as the activation function, which is defined as Equation (11).

$$ReLU(x) = max(0, x) = \begin{cases} x & x > 0 \\ 0 & x \leq 0 \end{cases} \tag{11}$$

The five neurons in the output layer correspond to the five base actions, and a Softmax normalization function is used to ensure that all actions sum to one, which is defined as shown in Equation (12).

$$softmax(X)_j = \frac{e^{X_j}}{\sum_{k=1}^{K} e^{X_k}} \tag{12}$$

where $softmax(X)_j$ denotes the value of the jth output under K outputs. The input of the policy network is its own state $S$, which is independent of the states of other agents. Each agent makes decisions through the strategy network, which depends on its own state $S$, thus reflecting the non-cooperative nature of the multi-agent interactive learning method.

The input layer of the critic network is all states $S$ and actions $A$ in the environment, and an agent can obtain information of all agents through observation, which obviously enhances the interactive learning ability of the multi-agent compared with the critic network of DDPG which only targets the states and actions of agents.

### 3.3.2. Multi-Agent Interactive Learning

The strategy parameters for each agent in multi-agent interactive learning are $\theta = (\theta_1, \theta_2, \ldots, \theta_N)$ and the joint strategy is $H: \pi = (\pi(\theta_1), \pi(\theta_2), \ldots, \pi(\theta_N))$

The multi-agent interactive learning experience replay set is

$$s_t, a_1^t, a_2^t, \ldots, a_N^t, r_1^t, r_2^t, \ldots, r_N^t, s_{t+1} \tag{13}$$

where $s_t = (o_1^t, o_2^t, \ldots, o_N^t)$ denotes the set of observations of an agent at moment t, $A_t = (a_1^t, a_2^t, \ldots, a_N^t)$ denotes the set of actions of an agent at moment t, $R_t = (r_1^t, r_2^t, \ldots, r_N^t)$ denotes the set of rewards obtained by the agent at moment t after performing their respective actions, and $s_{t+1} = (o_1^{t+1}, o_2^{t+1}, \ldots, o_N^{t+1})$ denotes the set of observations of the agent at moment t+1.

The critic network for multi-agent interactive learning contains the states and actions of all agents, and the evaluation is updated by observing the states and actions of other agents for interactive learning. The loss function of the critic network is

$$L = \frac{1}{K} \sum_{t=1}^{K} (y_t - Q(s_t, a_1, a_2, \ldots, a_N, \theta^Q))^2 \tag{14}$$

where $y_t$ is the target $Q$ value of the target critic network output, and the formula is shown in Equation (15). $Q(s_t, a_1, a_2, \ldots, a_N, \theta^Q)$ is the critic network output $Q$ value. The target $Q$ value and the critic network output $Q$ together update the critic network parameters

$$y_t = R_t + \gamma Q'(S_{t+1}, A_{t+1}, \theta^{Q'}) \tag{15}$$

where $A_t = \pi_{\theta^u}(S_t)$, and $\gamma$ is the discount factor.

The learning objective of the critic network is to minimize $L(\theta^Q)$, and the parameters $\theta^Q$ of the critic network are updated by the gradient descent method, and the formula is shown in Equation (4).

For the policy network, the objective function equation is shown in Equation (16).

$$J(\theta^u) = \frac{1}{K} \sum_{t=1}^{K} Q(s_t, a_1, a_2, \ldots, a_N, \theta^u) \tag{16}$$

The gradient calculation equation is

$$\nabla_{\theta^u} J = \frac{1}{K} \sum_{t=1}^{K} \nabla_{\theta^u} \pi(s_t, \theta^u) \nabla Q(s_t, a_1, a_2, \ldots, a_N, \theta^Q) \tag{17}$$

In Equation (17) above, except for $a_t$ corresponding to the current agent needs to be calculated in real time by using the policy network, the other actions can be obtained from the experience replay set. The goal of the policy network is to maximize the score of the critic network; thus, the parameters $\theta^u$ of the policy network can be updated by the gradient ascent method, as shown in Equation (7).

The multi-agent interactive learning framework is shown in Figure 6.

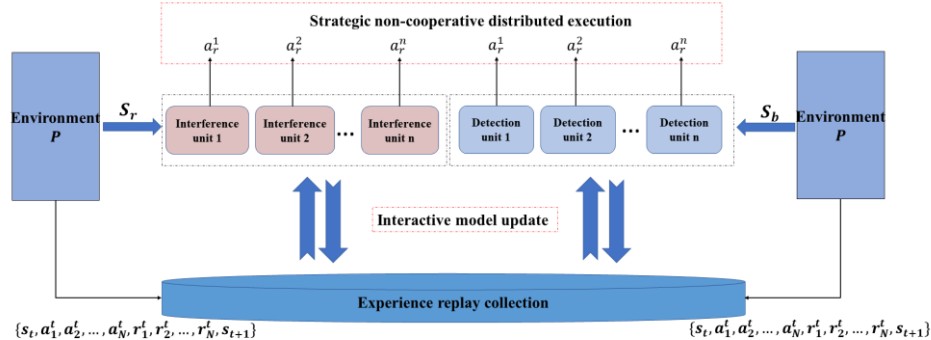

**Figure 6.** A framework for multi-agent interactive learning.

In Figure 6, the environment $P$ is interconnected and contains all agents. The policy network of each agent generates successive actions $a_t$ according to its own state $o_t$ in the environment, and the process is only related to the agent's own state; thus, the policy network reflects the non-cooperative nature of the multi-agent. The critic network is based on the state $o_t$ and the $a_t$ of the generated action, and the state $S_t$ and the $A_t$ of the generated action of the other agents in the experience playback set, and then the evaluation value $Q$ of the decision process of the policy network at moment t. Therefore, the critic network embodies the ability of multi-agent interactive learning.

The multi-agent interactive learning process is shown in Figure 7, and the pseudocode is shown in Table 1. During the training process, each agent relies on its own policy to obtain the action corresponding to the current moment state, and then executes the action to interact with the environment and obtains the experience and deposits it into the shared experience pool. The multi-agent interactive learning updates the local policy network by the global Q value, but it needs the global state information and the action information of all agents. The maximum number of training times are set to 25,000.

**Table 1.** Multi-agent interactive learning pseudocode.

---

**Multi-Agent Interactive Learning Pseudocode.**

---

1. Randomly initialize the network parameters $\theta^u$, $\theta^Q$, $\theta^{u'} = \theta^u$, $\theta^{Q'} = \theta^Q$, clear the experience playback set
2. For I from 1 to N，performing Iterations
   a) Initialize $S$ to the first state of the current state sequence.
   b) Each agent, at moment t, gets $A_t = \pi_\theta(S_t)$ in the strategy network based on state $S_t$
   c) Execute action $A_t$, get new state $S_{t+1}$, reward $R_t$, and store $\{s_t, a_1^t, a_2^t, \ldots, a_N^t, r_1^t, r_2^t, \ldots, r_N^t, s_{t+1}\}$ into experience playback set $D$
   d) Each agent starts updating the network by sampling K samples from the experience playback set $D$
      1) Compute the optimal action $A_{t+1} = \pi_\theta(S_{t+1})$ to be taken at the next moment through the target strategy network
      2) Approximate true $Q$ values are computed by the target evaluation network with state and action as inputs and $y_t = R_t + \gamma Q'(S_{t+1}, A_{t+1}, \theta^{Q'})$ as outputs
      3) Update the evaluation network with $L = \frac{1}{K}\sum_{t=1}^{K}(y_t - Q(s_t, a_1, a_2, \ldots, a_N, \theta^Q))^2$ as the loss function
      4) Update the policy network by$\nabla_{\theta_\pi}J = \frac{1}{K}\sum_{t=1}^{K}\nabla_{\theta_\pi}\pi(o, \theta^\pi)\nabla Q(s_t, a_1, a_2, \ldots, a_N, \theta^Q)$

---

5) If the number of iterations reaches the frequency of network parameter updates, the target evaluation network and target strategy network parameters are updated: $\theta^{Q\prime} = \tau\theta^Q + (1-\tau)\theta^{Q\prime}$

$$\theta^{u\prime} = \tau\theta^u + (1-\tau)\theta^{u\prime}$$ $\tau$ is the update coefficient, which is generally taken as 0.1 or 0.01

End for

End for

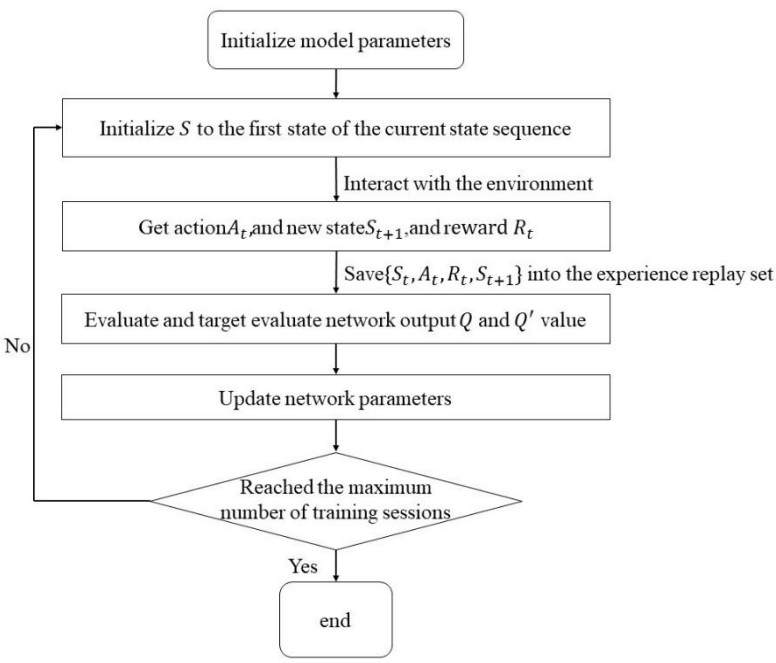

**Figure 7.** Multi-agent interactive learning flow chart.

### 4. Experimental Environment

In order to test the effect of the ISGE-NCE，three experiments are designed to be in different confrontation scenes with increasing complexity in conditions and with adding number of agents involved. The agents in the multi-agent are either interference source or detection source, which are marked in red or blue, respectively, for better illustration.

The distance between the blue detection source and the red interference source becomes smaller, the reward of the detection source increases. The distance between the red interference source and the blue detection source becomes larger, the reward of the interference source increases.

If the detection source collides with the interference source, which indicates that the capture of the detection source is effective, equivalently the interference source is subject to a capture. Otherwise, the red interference source gains reward by maintaining the distance to the blue detection source. The interactive self-learning and evolutionary capability of the ISGE-NCE method is evaluated by the reward curves and game results.

The initial parameters of the network $\theta^u$, $\theta^Q$ are set to 0.5, the state $S$ is the position of each agent, and the action $A$ is the velocity of the agent. The acceleration range of the experimental environment is set between 3.5 and 4.5; the maximum number of trainings is set to 25,000. The computer used for experiments whose cache is 16G, CPU is core i7, and operating system is Ubuntu 18.04. The code is implemented based on python3.6 with parl 1.3.1, Gym 0.10.5 and multi-agent particle environment.

### 4.1. A Basic Multi-Agent Confrontation Experiment(E1)

In the first experiment (which is named as E1 for short in the rest of this paper), the multi-agent confrontation environment contains two blue detection agents and two red interference agents.

Figure 8 presents the possible states of the agents in the process of multi-agent confrontation game and evolution.

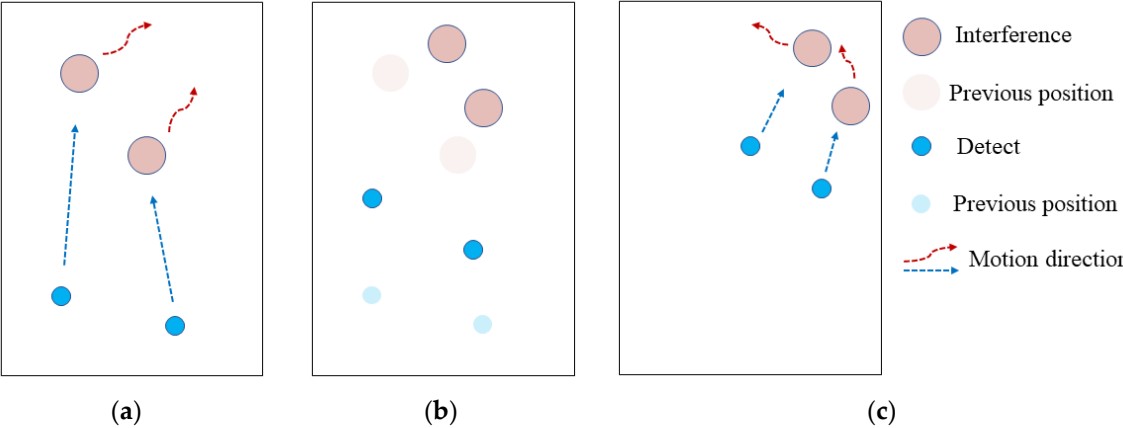

(**a**)        (**b**)        (**c**)

**Figure 8.** Confrontation process diagram of multi-agent confrontation environment. (**a**) Motion trend of the detection agent and the interference agent. (**b**) Action of the detection agent and the interference agent. (**c**) Action process of the detection agent and the interference agent.

After training, the agents would be aware of their goals and tasks. As shown in Figure 8a, the blue detects take the red interference as target, and their task is to chase the red interfering agent. The red interfering agent, targeting the blue detecting agent, is tasked to increase the distance to the blue detecting agent.

As shown in Figure 8b, the blue detection agent and the red interference agent act according to the task objective. Figure 8c shows that as the confrontation goes the blue detector agent keeps chasing the red interference agent, while the red interference agent can maintain the distance from the blue detection source.

The experimental results of the multi-agent confrontation environment show that the trained blue detection agent and red interference agent adversarial process is consistent with the expected effect of the initial reward model, and the adversarial parties can achieve interactive self-learning and evolution through game learning, which in turn validates the effectiveness of the ISGE-NCE method.

For further verification about the adaptability of the ISGE-NCE method for multi-agent confrontation under different environment, the acceleration of both adversaries was set to be inconstant. Denote $V_L$ as the acceleration of the blue detection agent and $V_R$ as that of the red interference agent. There would be three test conditions, $V_L > V_R$, $V_L = V_R$ and $V_L < V_R$, respectively. The values of the reward functions of both confrontation sides under different acceleration and different training times are presented in Table 2.

**Table 2.** Variation of reward function values under different acceleration conditions between two sides of the detection–interference confrontation.

| Training Times | Acceleration Conditions | | |
| --- | --- | --- | --- |
| | $V_L > V_R$ Rewards Change | $V_L = V_R$ Rewards Change | $V_L < V_R$ Rewards Change |
| 2000 | −32.2 | −24 | −19.16 |
| 10,000 | 14.19 | 10.25 | 14.01 |
| 15,000 | 30.24 | 19.18 | 12.55 |
| 20,000 | 73.5 | 27.48 | 17.62 |
| 25,000 | 71.88 | 25.5 | 13.22 |

As can be seen in Table 2, the rewards show an increase and eventually reach equilibrium with the training iterations. When the training goes over 2000 times, the reward function value starts to rise significantly.

A plot of the rewards recorded during the multi-agent confrontation is shown in Figure 9.

In Figure 9a, the reward value increases significantly at stage when the training goes between 13,000 and 17,000 times, and then becomes stable.

In Figure 9b, the rewards of the blue detectors increase significantly when the training goes between 7000 and 10,000 times, and then becomes stable. The similar situation happens for the red interferences, but the big variance of the rewards happens between the 3,000 and 5,000 training times, as shown in Figure 9c.

From Figure 9b,c, it is evident that both confrontation sides can significantly increase the reward and enhance the individual intelligence in the early stage of unsupervised training by using the ISGE-NCE method. The final almost steady reward states imply the evolution reaches the non-cooperative game equilibrium as expected.

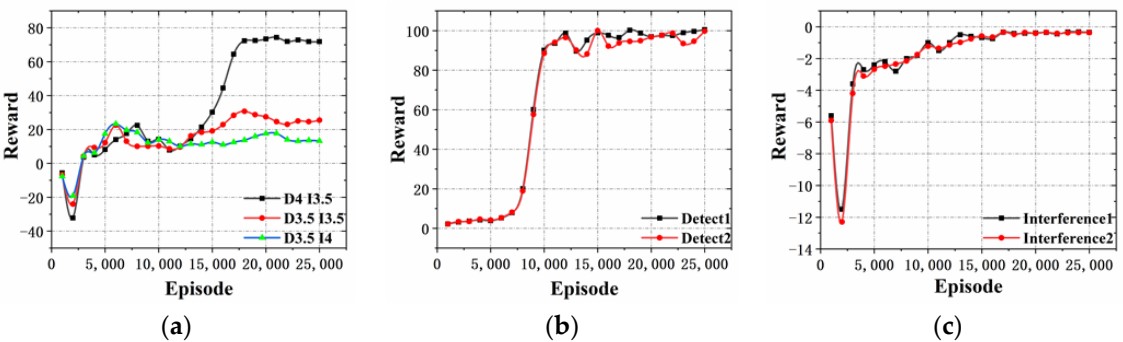

**Figure 9.** Reward function curves for multi-agent confrontation environment experiments: (**a**) reward function curves for the blue detection multi-agent and the red interference multi-agent for different acceleration parameters("D4,I3.5" indicates $V_L = 4$, $V_R = 3.5$; "D3.5,I3.5 " indicates $V_L = 3.5$ and $V_R = 3.5$; "D3.5,I4" indicates $V_L = 3.5$ and $V_R = 4$); (**b**) plot of the reward function for the blue detection agents, and (**c**) shows the plot of the reward function for the red interference agents.

### 4.2. Multi-Agent Covert Confrontation Environment (E2)

In order to evaluate the adaptability of the ISGE-NCE method for more complicated environments, a multi-agent covert confrontation environment experiment (which is called E2 for short) was designed based on the multi-agent confrontation environment experiment in Section 4.1, as shown in Figure 10.

Two covert regions are added, and the agent outside the covert region cannot obtain the position of the singleton which insides the covert region. The number of blue detection agents and red interference agents were both increased to 4.

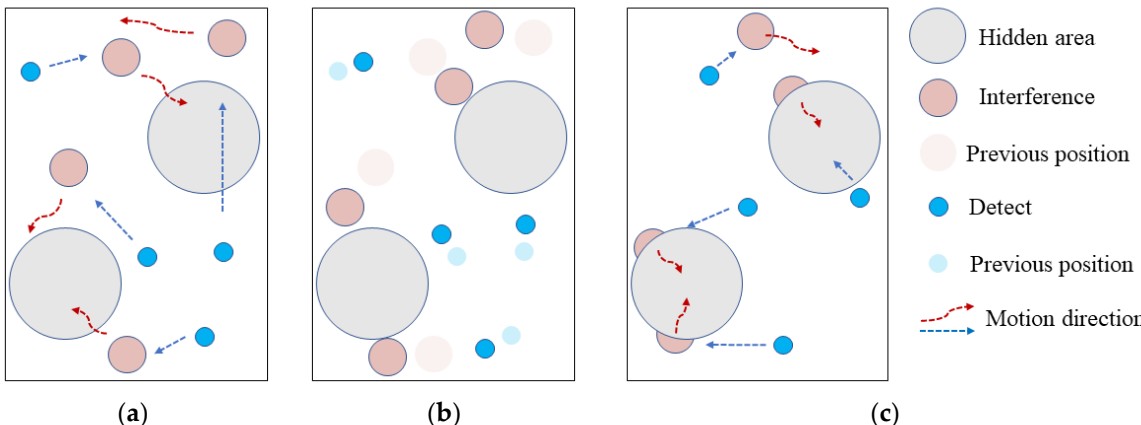

**Figure 10.** Multi-agent covert confrontation environmental confrontation process diagram: (**a**) motion trend of the detection agent and the interference agent; (**b**) action of the detection agent and the interference agent; (**c**) action process of the detection agent and the interference agent.

The behaviors of the blue detectors and the red interferences are basically similar with that was settled in Section 4.1. The changes are as following, in Figure 10, the blue detectors keep pursuing the red interfering agents while the red interference agents can maintain the distance from the blue detection source and use the environment to avoid it reasonably.

The results of the E2 show that the trained blue detection agents and red interference agents act as expected corresponding to the initial reward model, and the both confrontation sides can achieve interactive self-learning and evolution through game learning, thus verifying the effectiveness and adaptability of the ISGE-NCE method to complex environments.

Furthermore, different accelerations are set for both confrontation sides, which is similar with that have carried out in E1, to verify the performance of the ISGE-NCE method. The rewards of both confrontation sides under different acceleration and different training times are shown in Table 3, in which the $V_L$ and $V_R$ own the same meanings as introduced in E1, respectively.

**Table 3.** Variation of reward function values for different acceleration conditions on both confrontation sides of the detection–interference confrontation.

| Training Times | Acceleration Conditions | | |
|---|---|---|---|
| | $V_L > V_R$ Rewards Change | $V_L = V_R$ Rewards Change | $V_L < V_R$ Rewards Change |
| 2000 | −115.87 | −76.51 | −96.91 |
| 10,000 | 87.29 | 96.61 | 117.72 |
| 15,000 | 65.25 | 66.43 | 52.58 |
| 20,000 | 42 | 30.59 | 30.63 |
| 25,000 | 42.2 | 31.2 | 28.38 |

As can be seen from the training results in Table 3, the reward function values show an increase and eventually reach equilibrium with the training iterations. After the number of training times is greater than 2000, the reward function value starts to rise significantly. The reward function curves of the multi-agent confrontation environment experiment are shown in Figure 11. In Figure 11a, the reward value increases significantly between 3000 and 10,000 training times, and then stabilizes. In Figure 11b, the value of the reward function increases significantly between 2000 and 20,000 training times, and then stabilizes. In Figure 11c, the reward value increases significantly between 3000 and 5000

training times and then stabilizes. It shows that the both confrontation sides can significantly increase the value of the reward function and the degree of individual intelligence in the early stage of unsupervised training by the ISGE-NCE method, and then the curve stabilizes to reach the non-cooperative game equilibrium.

In Figure 11b,c, the blue detection agents and red interference agents evolve independently of each other in the interactive self-learning game and evolution and execute the non-cooperative distribution strategy. In addition, the reward function curves both finally reach the equilibrium steady state, further proving the strategy of non-cooperative equilibrium of the ISGE-NCE method.

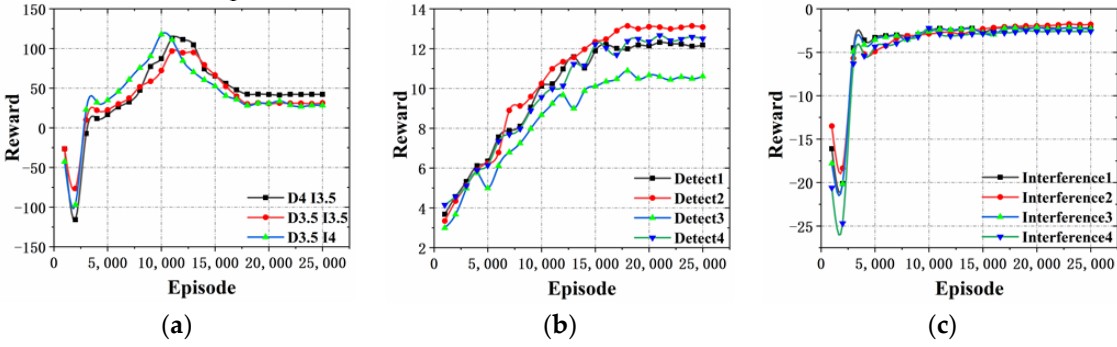

**Figure 11.** Experimental reward function curves for the multi-agent covert confrontation environment: (**a**) reward function curves for the blue detection multi-agent and the red interference multi-agent for different acceleration parameters; (**b**) curves of the reward function for the blue detection agents; (**c**) curves of the reward function for the red interference agents.

### 4.3. Multi-Agent Barrier-Covert Confrontation Environment (E3)

Based on the brilliant performance of our ISGE-NCE method in E1 and E2, the confrontation environment is updated to be more complicated in this Multi-agent Barrier-Covert Confrontation Environment (which is called E3 for short). As shown by Figure 12, a black barrier region is added as an impassable region, from which the agents would bounce back when they collide.

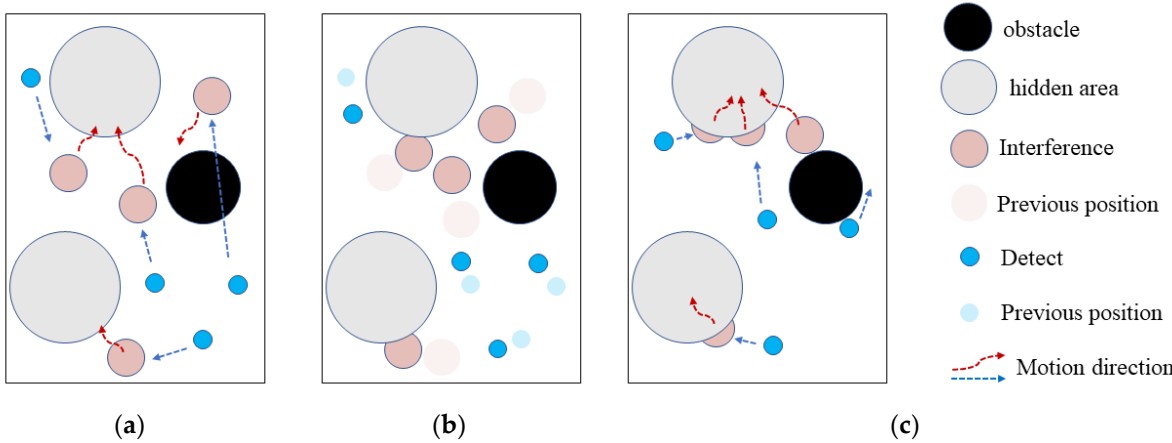

**Figure 12.** Diagram of the confrontation process in the multi-agent obstacle-covert confrontation environment: (**a**) detection agents' action, (**b**) interference agents' action, and (**c**) action process of the detection agent and the interference agent.

The multi-agent obstacle concealment confrontation environment experiment is shown in Figure 12, the blue detection agents keep pursuing the red interference agents, while the red interference agents can maintain the distance from the blue detection source and use the environment to avoid reasonably. The experimental results of the multi-agent confrontation environment show that the trained blue detection agents and red interference agents adversarial process is consistent with the expected effect of the initial reward

model, and the both confrontation sides can achieve interactive self-learning and evolution through game learning, thus verifying the effectiveness and adaptability of the ISGE-NCE method to complex environments.

Similarly, the rewards of the both confrontation sides under different acceleration and different training times are observed and presented in Table 4, to complement the performance of our ISGE-NCE method in E3.

**Table 4.** Variation of reward function values under different acceleration conditions between both confrontation sides of the detection- interference confrontation.

| Training Times | Acceleration Conditions | | |
| | $V_L > V_R$ Rewards Change | $V_L = V_R$ Rewards Change | $V_L < V_R$ Rewards Change |
|---|---|---|---|
| 2000 | −140.43 | −126.49 | −60.66 |
| 10,000 | 89.84 | 88.70 | 83.95 |
| 15,000 | 32.53 | 65.38 | 41.05 |
| 20,000 | 32.84 | 38.31 | 32.56 |
| 25,000 | 32.3 | 28.58 | 38.52 |

As can be seen from the training results in Table 4, the reward function values show an increase and eventually reach equilibrium with the training iterations. The reward function values start to rise significantly after the number of training times is greater than 2000. A curve of the reward function curve of the multi-agent confrontation environment experiment is shown in Figure 13. In Figure 13a, the reward value rises significantly between 3000 and 12,000 training times, and then stabilizes. In Figure 13b, the value of the reward function increases significantly between 2000 and 22,000 training times, and then stabilizes. In Figure 13c, the reward value increases significantly between 3000 and 5000 training times and then stabilizes. It shows that the both confrontation sides can significantly increase the value of the reward function and the degree of individual intelligence in the early stage of unsupervised training by the ISGE-NCE method, and then the curve stabilizes to reach the non-cooperative game equilibrium.

In Figure 13b,c, the blue detection agents and red interference agents evolve independently of each other in the interactive self-learning game and evolution, and they execute the non-cooperative distribution strategy. In addition, the reward function curves both finally reach the equilibrium steady state, further proving the strategy of non-cooperative equilibrium of the ISGE-NCE method.

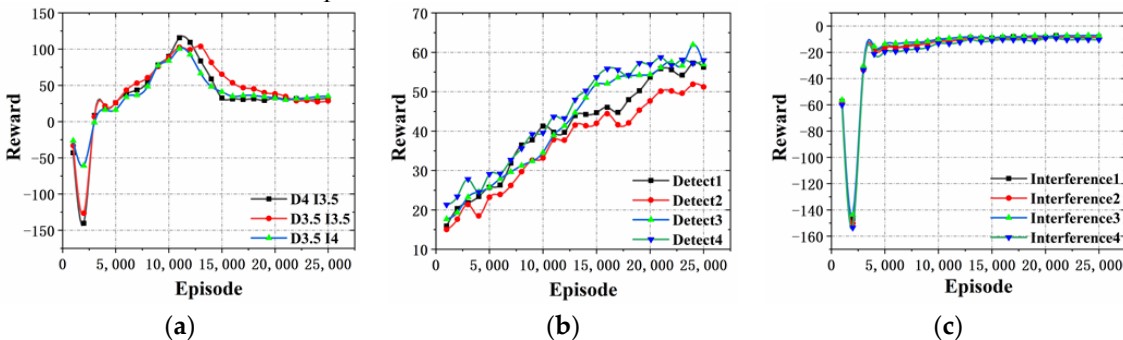

**Figure 13.** Experimental reward function curves for multi-agent barrier-covert confrontation environment: (**a**) reward function curves for the blue detection multi-agent and the red interference multi-agent with different acceleration parameters; (**b**) reward function plot for the blue detection agents; (**c**) reward function curves for the red interference agents.

### 4.4. Discussion of Experimental Results

#### 4.4.1. Learning Effect (Performance)

To assess the learning effect of our ISGE-NCE method, the behavior of the blue detection agents before and after learning, and the behavior of the red interference agents

before and after learning are compared in the three environmental experiments, i.e., E1, E2 and E3. It is assumed that the acceleration of blue detection agents and red interference agents are equal, and the number of collisions between them is considered as the evaluation index, and the upper limit of the number of collisions is set to 20.

Figure 14a shows a before-and-after training comparison for the red side, where the evolution proportion is the difference between the number of collisions before and after learning for the multi-agent. The evolution rate is the ratio of the evolution proportion to the upper limit of the number of collisions, which can indicate the degree of evolution of the individual intelligence. As Figure 14a can be obtained, the red interference multi-agent can successfully evade the blue detection multi-agent after learning, and its evolution rate reaches 60%. As can be obtained in Figure 14b, the blue detection multi-agent achieves accurate detection and pursuit of the red interference multi-agent through self-learning as well as antagonistic learning of the red interference multi-agent. The evolution rate reaches 80% compared to unlearning. The results of the effect plots before and after multi-agent learning show that the ability of both the red interference multi-agent and the blue detection multi-agent are effectively improved through multiple iterative learning, which further verifies the achievability of interactive self-learning game and evolution under the simulated confrontation environment experiment.

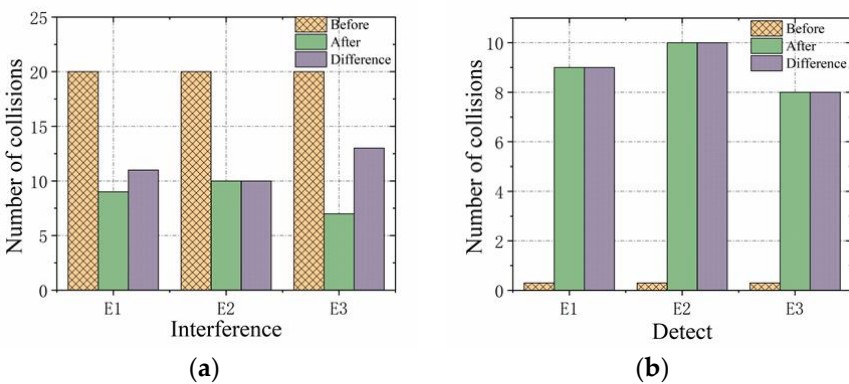

**Figure 14.** Comparison of the effect before and after multi-agent learning: (**a**) comparison of the effect before and after learning for the red interference multi-agent (orange represents the number of collisions before training, green represents the number of collisions after training, and purple represents the difference between the number of collisions before and after training); (**b**) comparison of the effect before and after learning for the blue detection multi-agent (orange represents the number of collisions before training, green represents the number of collisions after training, and purple represents the difference between the number of collisions before and after training).

### 4.4.2. Interactive Self-Learning and Evolutionary Effects (Performance)

As shown in Figure 15, the reward function values of interactive self-learning and evolution are compared with those of non-interactive self-learning and evolution in the three experimental environments settings of 4.1, 4.2 and 4.3. When the number of training times is 20,000, it can be seen in Figure 15a that the reward function value of E1 is 71.8, the reward function value of E2 is 42.3, and the reward function value of E3 is 32.4, while the reward function curves are close to equilibrium stability in the three experimental environments. The reduced learning time and the faster stabilization of the algorithm are attributed to the interactive learning ability of the critic network. In Figure 15b, the reward function values of E1 are 9.9, E2 are 8.2, and E3 are 6.8, while the reward function curves in the three experimental environments are obviously not converged; thus, it can be concluded that the ISGE-NCE can effectively improve the evolutionary ability and convergence.

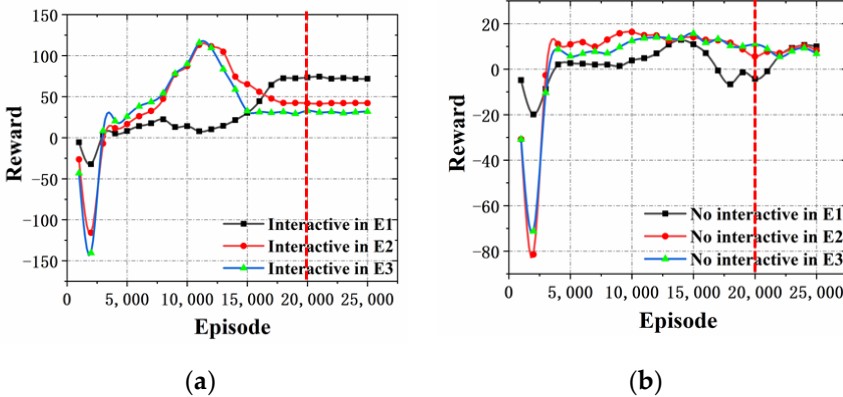

**Figure 15.** The reward function comparison chart of Interactive self-learning and evolutionary: (**a**) reward function plot of interactive self-learning and evolutionary; (**b**) reward function plot of no-interactive self-learning and evolutionary (E1 is the multi-agent confrontation environment, E2 is the multi-agent covert confrontation environment, and E3 is the multi-agent barrier-covert confrontation environment).

The number of training times for interactive self-learning and evolution when the reward function value reaches 10 for each experimental environments in 4.1, 4.2 and 4.3 is shown in Table 5. Table 5 indicates the number of training times required for each curve to reach a reward function value of 10. From Table 5, it can be obtained that ISGE-NCE can improve the rate of convergence by more than 46.3% compared with no interactive self-learning and evolution under the same reward function value. It further shows that introducing interactive learning framework into ISGE-NCE can reduce the time to reach non-cooperative equilibrium.

**Table 5.** Comparison of the training number of interactive self-learning and evolutionary.

| With or Without Interactive Training | Experimental Scene Categories | | |
|---|---|---|---|
| | E1 Rewards 10 Training Times | E2 Rewards 10 Training Times | E3 Rewards 10 Training Times |
| Yes | 6000 | 3000 | 4000 |
| No | 13,000 | 4000 | 10,000 |

### 4.4.3. Comparative Effectiveness with DDPG Learning Method

In order to verify the effectiveness of the non-cooperative equilibrium-based interactive self-learning and evolutionary methods, experiments were conducted using the DDPG learning method in three experimental environments introduced in Sections 4.1–4.3, and a comparison of the reward functions of the ISGE-NCE method and the DDPG learning method is shown in Figure 16. Based on the same reward model, the higher the reward value, the better the effect of detection and defense, which can reflect the better the performance of the method.

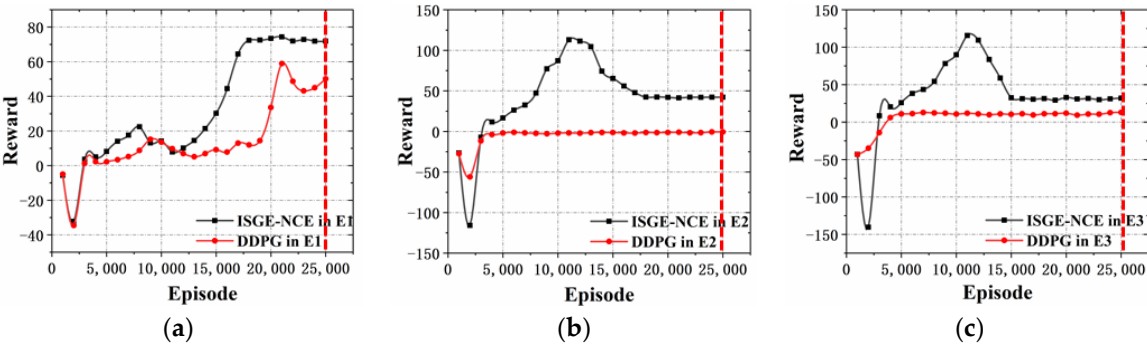

**Figure 16.** Comparison chart of the reward functions of the ISGE-NCE method-DDPG learning method: (**a**) comparison chart of the reward functions of the experimental environment (E1) in Section 4.1; (**b**) comparison chart of the reward functions of the experimental environment (E2) in Section 4.2; (**c**) comparison chart of the reward functions of the experimental environment (E3) in Section 4.3.

As shown in Figure 16a, the reward obtained using our ISGE-NCE method is 71.88 when the training goes around 25,000 times in E1(the experimental environment condition in Section 4.1), and the non-cooperative equilibrium has been reached after nearly 17,500 times' training. The reward by using the DDPG learning method is 50.12, and the equilibrium has not been reached by that time. This shows the ISGE-NCE method effectively improves the value of the reward function by more than 43.4% and converges faster. Additionally, as shown in Figure 16b, according to the results in E2, the reward using ISGE-NCE method is 42.2 while the reward using DDPG learning method is -0.6. It indicates that the reward is effectively improved by more than 50% by using our ISGE-NCE method compared with the DDPG learning method. Figure 16c shows that under the experimental environment conditions introduced in Section 4.3, the reward of ISGE-NCE method is 32.3 while the reward of DDPG learning method is 12.86. The reward is effectively improved by more than 20% by using our ISGE-NCE method. Lastly, when the number of agents is increased as explained in Sections 4.2 and 4.3, since the DDPG is an individual learning method which is not suitable for complex multi-agent environments, the training capacity is significantly constrained although the convergence speed is fast.

These results indicate that ISGE-NCE with better performance than DDPG. ISGE-NCE method can solve the problem of multi-agent learning without manual intervention, which can realize unsupervised learning training. ISGE-NCE has the characteristics of transfer learning, and the corresponding transfer tasks can be completed with a common evaluation index "reward". The reward model for different system tasks includes reward mechanism and observation module. The training data is stored in the "model" file after the task is completed and can be performed directly when the same task is encountered again. In summary, the ISGE-NCE method has a wide range of potential applications and is suitable for agent or multi-agent game confrontation environments.

## 5. Conclusions

In this paper, an interactive self-learning game and evolution approach based on non-cooperative equilibrium (ISGE-NCE) for multi-agent evolution was proposed. This method owns the advantages of both game theory and interactive self-learning, which is proven by three groups of multi-agent confrontation experiments. During the evolution experiments, the changing rewards before and after training under different conditions show that our ISGE-NCE method can significantly improve the rewards and refine the individual intelligence at an early stage of training, which then promotes stabilizing the training process and reaching the state of non-cooperative game equilibrium efficiently. In addition, the ISGE-NCE method does not increase the additional consumption compared with the DDPG method.

The high evolution rates of both the interference and detection agents support our ISGE-NCE method to improve the efficiency of learning evolution by more than 46.3%

against the non-interactive self-learning approach under the same reward requirement. Finally, the learning effectiveness of our ISGE-NCE method is compared with the DDPG method, and our method gives 43.4%, 50%, and 20% higher in rewards, respectively, in three different experimental environments. The presented results demonstrate the superiority of the ISGE-NCE method in multi-agent intelligence.

**Author Contributions:** Conceptualization, Y.L. and M.Z.; methodology, Y.L., M.Z. and S.W.; software, M.Z.; validation, Y.L., M.Z. and H.Z.; formal analysis, Y.L. and S.W.; investigation, M.Z. and H.Z.; resources, Y.L. and S.W.; data curation, M.Z.; writing—original draft preparation, M.Z. and S.W.; writing—review and editing, M.Z., Y.L. and S.W.; visualization, H.Z. and F.Y.; supervision, H.Z. and F.Y.; project administration, S.W. and F.Y.; funding acquisition, Y.L. All authors have read and agreed to the published version of the manuscript.

**Funding:** This research was supported by the National Natural Science Foundation of China, with grant nos. 61973308,62003350 and 62175258, and the Fundamental Research Funds for the Central Universities of China with grant no. 800015Z1160.

**Data Availability Statement:** The data presented in this study are available upon reasonable request from the corresponding author.

**Conflicts of Interest:** The authors declare no conflict of interest.

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
