# Peer review of "An Interactive Self-Learning Game and Evolutionary Approach Based on Non-Cooperative Equilibrium"

_electronics, doi:10.3390/electronics10232977_

Round 1
Reviewer 1 Report
Dear Authors,
Thanks for your interesting work.
Although the paper discusses an interesting topic, there are some comments below need to be considered to make the paper in a better form.
- Please go through written English and improve the language and check the grammar errors
- Please remove the word “we” from the article
- you made a good review discussing the related work, but you still need to clarify the weaknesses for every reference.
- Abstract should be completely rewritten. Statements should be formed effectively to exhibit the actual contribution of the paper. The last two lines of the abstract should talk about the outcome of the case study. Make it catchier in only 200 words.
- Introduction should include the contribution of this paper, as in current form there would not be any contribution listed. Refine it. Also, add motivation to write this paper.
- Define the acronym for the first time appeared in the paper and use the same throughout the paper. At many places you have multiple times define the acronym.
- Also include a tabular comparison in the related work where you must compare all the available literature with some parameters like pros, cons, etc.
- Reduce the contents in the conclusion with more clarity.
- All the references should be complete and uniform as per the guidelines of the Template.
- Rewrite the conclusion with more clarity. Remove the repeated phrases from the conclusion. It should be 200 words.
- Check the reference section, which is completely not as per the guidelines of the journal, every reference must contain the name of all authors, paper title, journal/conference name, volume, issue number/conference location, page numbers, and year. Update all the references very carefully and take the references from the plain text file or bib file of the corresponding publication site like IEEE, Wiley, Springer, ScienceDirect.
- Some more recent references must be included to strengthen the quality of your work and make it more in line with the latest trends in technology.
- There are many grammatical mistakes, please proofread the complete paper to correct them. Use Grammarly or any other software to improve the language of this paper.
Author Response
Dear Reviewer,
Manuscript ID: electronics-1468441
Title: An interactive self-learning game and evolutionary approach based on
non-cooperative equilibrium
Thank you for your comments concerning our manuscript. Those comments are all valuable and very helpful for revising and improving our paper, as well as the important guiding significance to our researches. We have studied comments carefully and summarized them into thirteen points. We have made correction point by point and highlighted it in red.
Reviewer’s comments and Response to comments:
(1)Please go through written English and improve the language and check the grammar errors
Response: Given your suggestions that make appropriate language modifications, we checked the full text again and revised in the manuscript.
(2)Please remove the word “we” from the article
Response: Thank you for your precious comments and advice, and we checked the full text and replaced the word "we" with appropriate writing.
(3)you made a good review discussing the related work, but you still need to clarify the weaknesses for every reference.
Response: We gratefully thank you for the precious time the reviewer spent making constructive comments. Based on your suggestions, we summarized the weaknesses of the references, at the end of each paragraph of the introduction.
We pointed out that DDPG has problems such as behavioral convergence failure and low training efficiency when dealing with multi-agent environment behavior problems, at the end of the first paragraph of the introduction.
These methods of the second paragraph lack the support of game theory was stated, at the end of the second paragraph.
These studies of the third paragraph didn’t introduce game strategies was stated, at the end of the third paragraph.
Research in the fourth paragraph didn’t consider the non-cooperative equilibrium between the two parties of the game.
In the fifth paragraph, we pointed out that none of the non-cooperative game equilibrium based deep learning methods collected above had introduced interactive self-learning simultaneously.
(4)Abstract should be completely rewritten. Statements should be formed effectively to exhibit the actual contribution of the paper. The last two lines of the abstract should talk about the outcome of the case study. Make it catchier in only 200 words.
Response: We are extremely grateful to reviewer for pointing out this problem. We rewrote the abstract to make it more effective in stating the contribution of the article and kept the number of words under 200.
(5)Introduction should include the contribution of this paper, as in current form there would not be any contribution listed. Refine it. Also, add motivation to write this paper.
Response: We are grateful for the suggestion. We have added the contribution and write motivation of this paper in the penultimate paragraph of the introduction.
This paper solved the problems of poor effect and slow learning speed of DDPG on multi-agent learning. What’s more, the concept of ' game evolution ' is successfully introduced into the interactive self-learning framework, and the multi-agent interactive self-learning game and evolution method was designed. Finally, the result of agent training achieved the effect of non-cooperative equilibrium. More details could be found in red font.
(6)Define the acronym for the first time appeared in the paper and use the same throughout the paper. At many places you have multiple times define the acronym.
Response: Thank you for your precious comments. We checked the full text and removed the duplicate definition of the acronym.
(7)Also include a tabular comparison in the related work where you must compare all the available literature with some parameters like pros, cons, etc.
Response: We are grateful for the suggestion. Due to the different method test scenarios in the literature, the inconsistent evaluation indicators of different methods. For example, the air combat platform used in Literature 6 is used as a simulation tool, and some of the aircraft's power parameters are used as evaluation indicators. The experimental scene designed in Literature 7 is for the learning and training of the robotic arm.
In order to solve the problem of inconsistent test scenarios for each method, we designed three more general scenarios, as E1, E2, and E3 in Section 4, and compared different methods in these scenarios. And at the end, we did experimental demonstration and data analysis to prove the better performance of our proposed ISGE-NCE method. The result shows that in the three different experiment scenarios, compared with the DDPG, our ISGE-NCE method improves the multi-agent evolution effectiveness by 43.4%, 50%, and 20%, respectively, with low training costs.
(8)Reduce the contents in the conclusion with more clarity.
Response: We deeply appreciate your suggestion. We rewrote the conclusion to make it clearer.
(9)All the references should be complete and uniform as per the guidelines of the Template.
Response: Thank you for the suggestion. We revised the references in accordance with the template to ensure consistency.
(10)Rewrite the conclusion with more clarity. Remove the repeated phrases from the conclusion. It should be 200 words.
Response: Thank you for underlining this deficiency. the conclusion was rewritten and the repeated phrases was removed to make the expression clearer.
(11)Check the reference section, which is completely not as per the guidelines of the journal, every reference must contain the name of all authors, paper title, journal/conference name, volume, issue number/conference location, page numbers, and year. Update all the references very carefully and take the references from the plain text file or bib file of the corresponding publication site like IEEE, Wiley, Springer, ScienceDirect.
Response: Thank you for underlining this deficiency. We revised the references in accordance with the template to ensure consistency.
(12)Some more recent references must be included to strengthen the quality of your work and make it more in line with the latest trends in technology.
Response: We agree with your suggestions, and we have updated some references based on your suggestions. We introduced some more recent articles into the references, and analyzed the differences between these methods and ours in the introduction. Specific details can be found in references 12,19, and 24.
(13)There are many grammatical mistakes, please proofread the complete paper to correct them. Use Grammarly or any other software to improve the language of this paper.
Response: Thank you for your careful review. We are very sorry for the grammatical mistakes in this manuscript. We checked the full text and corrected these grammatical mistakes.
We tried our best to improve the manuscript and made some changes to it. We sincerely thank you for your enthusiastic work and hope that the correction will be approved.
Thank you again for your comments and suggestions.
Kind regards,
Suyu Wang
E-Mail: [email protected]
Reviewer 2 Report
The paper is well written and well structured. However, there are some issues that the authors should take care of:
- At the end of related work, the authors should remind the reader how their work differentiates compared to the previous approaches.
- Although the authors made a brief description of related literature.
- The authors should check the taxonomies and communicate to the potential reader the characteristics of transfer learning and especially if and how their system can mitigate them. Maybe a subsection should fit that purpose.
- No technical details are given about the testbed environment, such as the physical machine used for the experiments.
- The authors should also point out the low overhead introduced by their methodology in the conclusion.
- There are many up-to-date theoretical studies on Machine Learning and well-established communities working on different theoretical aspects and techniques (e.g. make your network shallower by fewer layers, use less number of hidden units or use Weight Agnostic strategies, decrease regularization, etc). The authors must extend the explanation about the main differences between the current submission and the previous studies like https://www.mdpi.com/2079-9292/10/7/781, https://www.mdpi.com/2079-9292/10/7/768/pdf, https://ieeexplore.ieee.org/stamp/stamp.jsp?arnumber=9363125, etc. I would suggest a comparison study.
- The authors should probably provide more information about the proposed architecture. This is a major issue of the paper of how the authors have chosen this specific architecture for the proposed processing method, how it emerged and why the proposed architecture is the optimal solution.
- The second major issue of the paper is the explanation of the results, which are presented completely casually and without thorough analysis.
- Figures are small and apparently of low resolution. If the authors consider that provides important information, they should definitely enlarge it to be clear and legible.
- The discussion section could include some contribution to the international literature.
Author Response
Dear Reviewer,
Manuscript ID: electronics-1468441
Title: An interactive self-learning game and evolutionary approach based on
non-cooperative equilibrium
Thank you for your comments concerning our manuscript. Those comments are all valuable and very helpful for revising and improving our paper, as well as the important guiding significance to our researches. We have studied comments carefully and summarized them into ten points. We have made correction point by point and highlighted it in red.
Reviewer’s comments and Response to comments:
(1)At the end of related work, the authors should remind the reader how their work differentiates compared to the previous approaches.
Response: We deeply appreciate the reviewer’s suggestion. According to the reviewer’s comment, we added the difference between the method in this article and the DDPG method at the end of the related work. More details could be found in red font.
Since the network update equation of the DDPG learning method does not con-sider the states and actions of other agents, it cannot achieve interactive self-learning for game evolution. For complex multi-agent environments, the training results of the DDPG learning method are difficult to converge, and it is also difficult to achieve a non-cooperative equilibrium. In order to solve this problem, this paper proposes an ISGE-NCE method. Generative adversarial networks and group interactive learning framework are designed to overcome the shortcomings of DDPG method.
(2)Although the authors made a brief description of related literature.
Response: We gratefully thank you for the precious time the reviewer spent making constructive comments. Based on your suggestions, we summarized the weaknesses of the references, at the end of each paragraph of the introduction.
We pointed out that DDPG has problems such as behavioral convergence failure and low training efficiency when dealing with multi-agent environment behavior problems, at the end of the first paragraph of the introduction.
These methods of the second paragraph lack the support of game theory was stated, at the end of the second paragraph.
These studies of the third paragraph didn’t introduce game strategies was stated, at the end of the third paragraph.
Research in the fourth paragraph didn’t consider the non-cooperative equilibrium between the two parties of the game.
In the fifth paragraph, we pointed out that none of the non-cooperative game equilibrium based deep learning methods collected above had introduced interactive self-learning simultaneously.
(3)The authors should check the taxonomies and communicate to the potential reader the characteristics of transfer learning and especially if and how their system can mitigate them. Maybe a subsection should fit that purpose.
Response: Thank you for your precious advice which are very helpful for revising and improving our paper. In discussion section, we stated the transfer learning characteristics of our method and how to adapt this method to other systems to facilitate the reading of potential readers.
ISGE-NCE has the characteristics of transfer learning, and the corresponding transfer tasks can be completed with a common evaluation index ‘reward’. The reward model for different system tasks includes reward mechanism and observation module. The training data is stored in the ‘model’ file after the task is completed and can be done directly when the same task is encountered again. In summary, the ISGE-NCE method has a wide range of potential applications and is suitable for agent or multi-agent game confrontation environments. More details could be found in the red font in discussion section.
(4)No technical details are given about the testbed environment, such as the physical machine used for the experiments.
Response: Thank you for underlining this deficiency. We gave technical details of the testbed environment in section 4.
The computer used for experiments whose cache is 16G, CPU is core i7, and operating system is Ubuntu 18.04. The code is implemented based on python3.6 with parl 1.3.1, Gym 0.10.5 and multi-agent particle environment.
(5)The authors should also point out the low overhead introduced by their methodology in the conclusion.
Response: We are extremely grateful to reviewer for pointing out this problem. We stated the low consumption of the method and other characteristics in the conclusion.
The ISGE-NCE method does not increase the additional consumption compared with the DDPG method, so it does not affect the operating speed of the experimental equipment
(6)There are many up-to-date theoretical studies on Machine Learning and well-established communities working on different theoretical aspects and techniques (e.g. make your network shallower by fewer layers, use less number of hidden units or use Weight Agnostic strategies, decrease regularization, etc). The authors must extend the explanation about the main differences between the current submission and the previous studies like https://www.mdpi.com/2079-9292/10/7/781, https://www.mdpi.com/2079-9292/10/7/768/pdf, https://ieeexplore.ieee.org/stamp/stamp.jsp?arnumber=9363125, etc. I would suggest a comparison study.
Response: We deeply appreciate the reviewer’s suggestion. We have read these and other articles carefully and benefited a lot from them. In addition, we introduced these articles into the references, and analyzed the differences between these methods and ours in the introduction. Specific details can be found in references 12,19, and 24.
(7)The authors should probably provide more information about the proposed architecture. This is a major issue of the paper of how the authors have chosen this specific architecture for the proposed processing method, how it emerged and why the proposed architecture is the optimal solution.
Response: Thank you for your comment, and our reply is as follows: ISGE-NCE system overcomes the incompetence of DDPG 's long training time and poor effect on multi-agent learning. The generative adversarial network part realizes the rapid identification of monomer categories to improves the training speed. And the interactive learning framework part enhances training capacity. In addition, we have added more information about this architecture in section 3.1.
Generative adversarial networks can quickly classify the individual under certain evolution environment. There are agents different in category in the environment set, and the generative model continuously generates new agents by introducing hidden random variables. The discriminant model takes the form of a centralized sampling to distinguish between concrete classes of both sides. Both the generative and discriminative models are updated using back propagation to achieve a stable equilibrium. The generative adversarial network lays the foundation for multi-agent interactive learning and helps to improve the interactive recognition capability and classification speed of the system. Based on the common rules, the multi-agent interactive learning has an additional parameter sharing mechanism compared with DDPG learning. In addition, the critic network would consider the state and action parameters of other agents, which enhances the interactive learning ability of the agents and speed up the training process as well. Besides, the game evolution of the multi-agent would promote the learning efforts and improve the learning efficiency. Training results reflect non-cooperative equilibrium.
(8)The second major issue of the paper is the explanation of the results, which are presented completely casually and without thorough analysis.
Response: Thank you for underlining this deficiency. We analyzed the result data in depth in Section 4.4 and explained the result indicators in detail. In addition, we explained the relationship between the number of training times and the value of the return function and the method in this article, as well as the reason why it represents the convergence speed and performance. More details could be found in the red font in discussion section.
(9)Figures are small and apparently of low resolution. If the authors consider that provides important information, they should definitely enlarge it to be clear and legible.
Response: Thank you for your careful review. We have re-adjusted the resolution of some pictures and bolded the fonts in the pictures to ensure that readers can read better.
(10)The discussion section could include some contribution to the international literature.
Response: We deeply appreciate the reviewer’s suggestion. In discussion section, we emphasized the significance of the ISGE-NCE method, as well as its low loss, fast training, and high performance, which can solve complex group evolution problems. We indicate that ISGE-NCE with better performance than DDPG. IS-GE-NCE method can solve the problem of multi-agent learning without manual intervention, which can realize unsupervised learning training. More details could be found in the red font in discussion section.
We tried our best to improve the manuscript and made some changes to it. We sincerely thank you for your enthusiastic work and hope that the correction will be approved.
Thank you again for your comments and suggestions.
Kind regards,
Suyu Wang
E-Mail: [email protected]

Round 2
Reviewer 1 Report
can be accepted
Reviewer 2 Report
Based on the comments raised by the examiners all of the doubtful and blurred areas have been well explained by the authors. Now it is ready for publication.